# Phosphine-Functionalized Core-Crosslinked Micelles and Nanogels with an Anionic Poly(styrenesulfonate) Shell: Synthesis, Rhodium(I) Coordination and Aqueous Biphasic Hydrogenation Catalysis

**DOI:** 10.3390/polym14224937

**Published:** 2022-11-15

**Authors:** Hui Wang, Chantal J. Abou-Fayssal, Christophe Fliedel, Eric Manoury, Rinaldo Poli

**Affiliations:** 1CNRS, LCC (Laboratoire de Chimie de Coordination), Université de Toulouse, UPS, INPT, 205 Route de Narbonne, BP 44099, 31077 Toulouse CEDEX 4, France; 2Centre for Catalysis and Sustainable Chemistry, Department of Chemistry, Technical University of Denmark, Kemitorvet, Building 207, 2800 Kongens Lyngby, Denmark; 3Institut Universitaire de France, 1, Rue Descartes, 75231 Paris CEDEX 05, France

**Keywords:** aqueous biphasic catalysis, rhodium, hydrogenation, core-crosslinked micelles, poly(styrenesulfonate), RAFT polymerization, polymerization-induced self-assembly

## Abstract

Stable latexes containing unimolecular amphiphilic core-shell star-block polymers with a triphenylphosphine(TPP)-functionalized hydrophobic core and an outer hydrophilic shell based on anionic styrenesulfonate monomers have been synthesized in a convergent three-step strategy by reversible addition-fragmentation chain-transfer (RAFT) polymerization, loaded with [RhCl(COD)]_2_ and applied to the aqueous biphasic hydrogenation of styrene. When the outer shell contains sodium styrenesulfonate homopolymer blocks, treatment with a toluene solution of [RhCl(COD)]_2_ led to undesired polymer coagulation. Investigation of the interactions of [RhCl(COD)]_2_ and [RhCl(COD)(PPh_3_)] with smaller structural models of the polymer shell functions, namely sodium *p*-toluenesulfonate, sodium styrenesulfonate, and a poly(sodium styrenesulfonate) homopolymer in a biphasic toluene/water medium points to the presence of equilibrated Rh-sulfonate interactions as the cause of coagulation by inter-particle cross-linking. Modification of the hydrophilic shell to a statistical copolymer of sodium styrenesulfonate and poly(ethylene oxide) methyl ether methacrylate (PEOMA) in a 20:80 ratio allowed particle loading with the generation of core-anchored [RhCl(COD)TPP] complexes. These Rh-loaded latexes efficiently catalyze the aqueous biphasic hydrogenation of neat styrene as a benchmark reaction. The catalytic phase could be recovered and recycled, although the performances in terms of catalyst leaching and activity evolution during recycles are inferior to those of equivalent nanoreactors based on neutral or polycationic outer shells.

## 1. Introduction

Aqueous biphasic catalysis is an elegant solution to facilitate catalyst recovery and recycling for processes that involve hydrophobic reactants and products [1], particularly when the catalyst is very expensive. To avoid catalyst leaching, the catalyst should have the lowest possible solubility in the reactant/product phase, while a rapid phase separation at the end of the reaction is critical for process efficiency. In addition, the transformation should ideally not suffer from mass transport limitations. Since the catalyst must not enter the reactant/product phase, the catalytic act must occur in the aqueous phase, requiring non-zero water solubility for the reactants. The Rh/TPPTS-catalyzed hydroformylation of propene and butene (TPPTS = triphenylphosphine trisulfonate) is an example of an industrially implemented, large-scale aqueous biphasic process that obeys these principles [2]. This process, however, is inefficient for higher olefins, which do not have sufficient water solubility. Several approaches have been investigated to remove this bottleneck, including catalyst anchoring on thermomorphic polymers [3], adding molecular transporters such as cyclodextrines [4,5], or increasing the water/organic interface through the addition of surfactants [6,7].

We have been interested in a different approach based on the use of micellar nanoreactors that contain a hydrophobic core and a hydrophilic shell. The core serves to anchor the catalyst and constitutes a suitable medium for the reactants, while the shell stabilizes the nanoreactors in the aqueous phase dispersion (latex) [8,9,10]. The use of amphiphilic diblock copolymers is particularly attractive because their self-assembly provides kinetically stable micelles in water. Additional advantages can be obtained by cross-linking the micellar arms, either at the shell or at the core level, thus generating unimolecular nanoreactors [11,12,13,14]. This modification removes the potential problems of extensive swelling and equilibria with free single chains, which are stabilized at the water-organic interface, and inverted micelles, which are stabilized in the organic phase, leading respectively to slow decantation and catalyst leaching.

Our approach is based on a one-pot synthesis of nanoreactors with a crosslinked core by a convergent reversible addition-fragmentation chain-transfer (RAFT) polymerization [15,16], which produces an aqueous phase dispersion of the nanoreactors that can be directly used in catalysis. Two different polymer architectures, named core-crosslinked micelles (CCM) and nanogels (NG), see Figure 1, were obtained by a small modification of the polymerization strategy. For the CCM synthesis, the water-soluble block resulting from the first step is chain-extended with only linear monomers, including the ligand-functionalized monomer, to yield a ligand-functionalized linear amphiphilic diblock copolymer. This step starts as a dispersion polymerization and entails “polymerization-induced self-assembly” (PISA) [17,18,19,20] to generate micelles, after which it continues as an emulsion polymerization. A cross-linker is then used to extend the amphiphilic diblock chains in the third and final step, yielding a unimolecular micelle with a small nanogel core. For the NG synthesis, on the other hand, all hydrophobic monomers (linear ones and cross-linker) are copolymerized together in the second and final step to chain-extend the water-soluble precursor. The main difference between the two architectures is that the ligands are anchored on flexible arms outside the crosslinked area in the CCM, whereas they are incorporated within the crosslinked area in the NG.

First generation CCM and NG polymers were built with a neutral hydrophilic shell based on linear chains of randomly copolymerized methacrylic acid (MAA) and poly(ethylene oxide) methyl ether methacrylate (PEOMA), where the PEO chains have an average degree of polymerization of 19, a functionalized polystyrene-based (PSt) core, and diethylene glycol dimethacrylate (DEGDMA) was used as a cross-linker [21,22,23,24,25,26,27,28]. The desired ligand was introduced by copolymerization of a suitably functionalized styrene in the second step, mostly 4-diphenylphosphinostyrene (DPPS), to yield PSt-anchored triphenylphosphine (TPP) functions [21,25], but nanoreactors with other PSt-anchored ligands such as bis(*p*-methoxyphenyl)phenylphosphine [23], nixantphos [29], and a Rh-coordinated N-heterocyclic carbene [30] have also been developed. After the coordination of suitable pre catalysts, these polymers proved efficient and recyclable in aqueous biphasic 1-octene hydroformylation [21,22,23,25] and in 1-octene and styrene hydrogenation [27]. However, the phase separation after catalysis turned out rather slow and led to non-negligible Rh leaching (ppm level) in the organic phase. The Rh leaching was shown to be associated to the transfer of the full nanoreactors to the organic phase, not to the loss of molecular Rh complexes from the nanoreactor core [22,25], which is caused by the increased lipophilicity of the PEOMA grafts in the outer shell at the higher temperatures used in catalysis. Irreversible nanoreactor agglomeration, caused by interpenetration of the polymer particles and subsequent cross-linking though Rh-P binding, also limited the efficiency of these nanoreactors [26].

In order to correct these problems, second-generation nanoreactors with an outer shell based on quaternized (methylated) 4-vinylpyridine (4VP) monomers, -[CH_2_-CH(4-C_5_H_4_NMe^+^I^−^)]_n−_, P(4VPMe^+^I^−^), were then developed, as seen in Figure 1. After optimizing the polymer synthesis [31], CCM and NG particles with a TPP-functionalized PSt core were generated, charged with [RhCl(COD)]_2_ to yield a core-anchored [RhCl(COD)(TPP)] precatalyst, and applied to the aqueous biphasic hydrogenation of styrene and 1-octene [32]. As anticipated, these catalytic nanoreactors exhibited improved properties: much faster decantation rates (minutes rather than hours) and lower catalyst leaching (<100 ppb rather than >1 ppm), while the turnover frequency (TOF) was similar to that of the first-generation nanoreactors, indicating equally efficient mass transport of the organic reactants and products through the neutral and polycationic shells.

We now wish to report the synthesis and catalytic application of third-generation nanoreactors containing a permanently charged polyanionic outer shell built with poly(sodium styrenesulfonate) chains. This new type of outer shell is also able, thanks to the Coulombic repulsion, to block particle interpenetration and is not characterized by high-temperature hydrophobicity. The investigation of new types of shells was motivated by the quest for further performance improvements in terms of decantation speed and catalyst leaching. In addition, the successful strategy for the fabrication of the 2nd-generation nanoreactor was rather tedious [31,32]. While these 3rd-generation polymers were more easily accessible than the 2nd generation, they presented an unexpected complication at the catalyst loading stage. Model studies with molecular compounds have allowed us to establish the reasons for these complications and to find a suitable solution through further shell modification, leading to the development of functional catalytic nanoreactors.

## 2. Materials and Methods

All manipulations were performed by Schlenk-line techniques under an inert atmosphere of dry argon. Solvents were dried by standard procedures and distilled under argon prior to use. 4,4′-azobis(4-cyanopentanoic acid) (ACPA, >98%, Fluka), sodium 4-vinylbenzenesulfonate (SS^−^Na^+^, >90%, Aldrich), poly(ethylene oxide) dimethyl ether (PEOMA, M_n_ = 950 g mol^−1^, Sigma-Aldrich), diethylene glycol dimethacrylate (DEGDMA, 95%, Aldrich), and 1,3,5-trioxane (>99%, Aldrich), were used as received. Styrene (St, 99%, Acros) was distilled under reduced pressure prior to use. The RAFT agent 4-cyano-4-thiothiopropylsulfanyl pentanoic acid (CTPPA) or R_0_-SC(S)S*n*Pr (R_0_ = -C(CH_3_)-(CN)-CH_2_CH_2_COOH), was prepared according to the literature [33]. The R_0_-(SS^−^Na^+^)_140_-SC(S)S*n*Pr and R_0_-(SS^−^Na^+^)_140_-*b-*St_50_-SC(S)S*n*Pr macroRAFT agents were synthesized as described in our recent contribution [34]. The deionized water used for the syntheses and DLS analyses was obtained from a Purelab Classic UV system (Elga Lab-Water).

### 2.1. Characterization Techniques

#### 2.1.1. NMR Spectroscopy

All nuclear magnetic resonance spectra were recorded in 5 mm diameter tubes at 297 K on a Bruker Avance 400 spectrometer. The ^1^H and ^31^P chemical shifts were determined using the residual peak of the deuterated solvent (δ 2.50 for DMSO-*d*_6_, 4.79 for D_2_O) as internal standard and are reported in ppm (δ) relative to tetramethylsilane. The monomer conversions in the polymerizations were monitored by ^1^H NMR in DMSO-*d*_6_ at room temperature by the relative integration of the protons of the internal reference (1,3,5-trioxane) at 5.20 ppm and the vinylic monomer protons.

#### 2.1.2. Size Exclusion Chromatography (SEC)

The P(SS^−^Na^+^) chain growth was monitored by SEC in water/acetonitrile (80/20 *v*/*v*) with 0.1 M NaNO_3_ at 60 °C at a flow rate of 1.0 mL min^−1^ by using a Viscotek TDA305 apparatus (SEC-DMF) coupled with a multi-angle light scattering (MALLS) detector (18 angles) from MALLS Wyatt Dawn Heleos. The measured d*n*/d*c* values for the R_0_-[(SS^−^Na^+^)_0.2_-*co*-PEOMA_0.8_]_50_-SC(S)S*n*Pr in the eluent is 0.051 mL g^−1^. All polymers were analyzed at a concentration of 5 mg mL^−1^ after filtration through a 0.45 μm pore size membrane. The separation was carried out on two columns from Agilent Aquagel OH Mixed M. The software used for data collection and calculation was OmniSec version 4.7 from Malvern Instruments.

#### 2.1.3. Dynamic Light Scattering (DLS)

The intensity-average diameters of the latex particles (*D*_z_) and the polydispersity index (PDI) were obtained on a Malvern Zetasizer NanoZS equipped with a He-Ne laser (λ = 633 nm), operating at 25 °C. Samples were analyzed after dilution (with deionized water) either unfiltered or after filtration through a 0.45 μm pore-size membrane. The procedure without filtration allowed verification of the presence of agglomerates. Zeta potential (ζ) determinations were also conducted on the same instrument by measuring the electrophoretic mobility.

#### 2.1.4. Transmission Electron Microscopy (TEM)

The morphological analyses of the copolymer nano-objects were performed at the Centre de Microcaractérisation Raimond Castaing (Toulouse France) with a JEOL JEM 1400 transmission electron microscope working at 120 kV. Diluted latex samples were dropped on a formvar/carbon-coated copper grid and dried under vacuum for 24 h.

### 2.2. Synthesis of Phosphine-Functionalized Copolymer Nanoreactors with an Anionic P(SS^−^Na^+^) Shell

*Preparation of latexes of the R_0_-(SS^−^Na^+^)_140_-b-St_50_-b-(St_1-y_-co-DPPS_y_)_300_-SC(S)SnPr amphiphilic triblock copolymers*. The synthesis of all latexes of this type followed the same procedure, which is detailed here only for the product with *y* = 0.1. To the pale-yellow latex of the R_0_-(SS^−^Na^+^)_140_-*b-*St_50_-SC(S)S*n*Pr macroRAFT agent (0.04 mmol of polymer, corresponding to 0.56 mmol of SS^−^Na^+^ units, 3 mL of water) in a Schlenk tube under Ar were added degassed styrene (1.24 mL, 1.12 g, 10.75 mmol; 270 equiv. per chain), DPPS (0.344 g, 1.20 mmol; 30 equiv. per chain), and trioxane (ca. 90 mg) as an internal standard. A portion of a degassed ACPA/NaHCO_3_ stock solution (0.11 mL, 2.2 mg ACPA, 7.96 µmol) was then added and the resulting reaction mixture was stirred at 80 °C for 7 h, yielding a white opalescent stable dispersion. The resulting polymer has a theoretical molar mass of 71,099 g mol^−1^. The weight percent polymer in the latex is 18.3%. Using the same amounts of R_0_-(SS^–^Na^+^)_140_-*b-*St_50_-SC(S)S*n*Pr, ACPA solution, water and trioxane but different amounts of degassed styrene and DPPS led to latexes of the product with different molar DPPS content (5% and 20%) in the hydrophobic block. For *y* = 0.05: R_0_-(SS^−^Na^+^)_140_-*b-*St_50_-SC(S)S*n*Pr (0.04 mmol of polymer, 3 mL of water), styrene (1.30 mL, 1.18 g, 11.34 mmol; 285 equiv. per chain), DPPS (0.17 g, 0.60 mmol; 15 equiv. per chain), M_n,th_ = 68,312 g mol^−1^, polymer content = 17.7% (*w*/*w*). For *y* = 0.2: R_0_-(SS^−^Na+)_140_-*b-*St_50_-SC(S)S*n*Pr (0.04 mmol of polymer, 3 mL of water), styrene (1.10 mL, 1.00 g, 9.55 mmol; 240 equiv. per chain), DPPS (0.69 g, 2.39 mmol; 60 equiv. per chain), M_n,th_ = 76,582 g mol^−1^, polymer content = 19.5% (*w*/*w*).

*Preparation of latexes of the R_0_-(SS^−^Na^+^)_140_-b-(St_1-y_-co-DPPS_y_)_300_-SC(S)SnPr amphiphilic diblock copolymers*. The synthesis of all latexes of R_0_-(SS^−^Na^+^)_140_-*b*-(S_1-*y*_-*co*-DPPS*_y_*)_300_-SC(S)S*n*Pr (*y* = 0.05, 0.1, 0.2, 0.25) followed the same procedure as for the synthesis of R_0_-(SS^−^Na^+^)_140_-*b*-St_50_-*b-*(St_1-*y*_-*co*-DPPS*_y_*)_300_-SC(S)S*n*Pr described in the previous section. For *y* = 0.05: R_0_-(SS^−^Na^+^)_140_-SC(S)S*n*Pr (0.04 mmol of polymer, 3 mL of water), styrene (1.31 mL, 1.19 g, 11.4 mmol; 285 equiv. per chain), DPPS (0.17 g, 0.6 mmol; 15 equiv. per chain), M_n,th_ = 63,203 g mol^−1^, polymer content = 16.7% (*w*/*w*). For *y* = 0.1: R_0−_(SS^−^Na^+^)_140_-SC(S)S*n*Pr (0.04 mmol of polymer, 6 mL of water), styrene (1.24 mL, 1.12 g, 10.7 mmol; 270 equiv. per chain), DPPS (0.34 g, 1.20 mmol; 30 equiv. per chain), M_n,th_ = 65,834 g mol^−1^, polymer content = 14.4% (*w*/*w*). For *y* = 0.2: R_0_-(SS^−^Na^+^)_140_-SC(S)S*n*Pr (0.04 mmol of polymer, 6 mL of water), styrene (1.1 mL, 1.00 g, 9.6 mmol; 240 equiv. per chain), DPPS (0.69 g, 2.4 mmol; 60 equiv. per chain), M_n,th_ = 71,347 g mol^−1^, polymer content = 15.5% (*w*/*w*). For *y* = 0.25: R_0_-(SS^−^Na^+^)_140_-SC(S)S*n*Pr (0.04 mmol of polymer, 6 mL of water), styrene (1.03 mL, 0.94 g, 9.0 mmol; 225 equiv. per chain), DPPS (0.86 g, 3.0 mmol; 75 equiv. per chain), M_n,th_ = 74,104 g mol^−1^, polymer content = 16.0% (*w*/*w*).

*Preparation of CCMs with a 90:10 St/DEGDMA core: cross-linking of the R_0_-(SS^−^Na^+^)_140_-b-St_50_-b-(St_1-y_-co-DPPS_y_)_300_-SC(S)SnPr amphiphilic block copolymers*. To the total volume of each of the R_0_-(SS^−^Na^+^)_140_-*b*-St_50_-*b*-(St_1-*y*_-*co*-DPPS*_y_*)_300_-SC(S)S*n*Pr latexes obtained as described above (0.04 mmol of polymer) were successively added DEGDMA (0.133 mL, 144.3 mg, 0.6 mmol; 15 equiv. per chain), styrene (0.62 mL, 561.6 mg, 5.4 mmol; 135 equiv. per chain) and the degassed ACPA/NaHCO_3_ stock solution (0.11 mL, 2.2 mg ACPA, 7.96 µmol). The resulting reaction mixtures were stirred at 80 °C for 4 h resulting in complete comonomer consumption (^1^H NMR monitoring in DMSO-*d*_6_) to yield the CCMs R_0_-(SS^−^Na^+^)_140_-*b*-(St_1-y_-*co*-DPPS*_y_*)_300_-*b*-(DEGDMA_0.1_-*co*-St_0.9_)_150_-SC(S)S*n*Pr (*y* = 0.05, 0.1, 0.2). Polymer content: 21.2% (*y* = 0.05), 21.7% (*y* = 0.1), 22.8% (*y* = 0.2); [TPP] = 38.22 μmol mL^−1^ (*x* = 0.05), 76.90 μmol mL^−1^ (*x* = 0.1), 154.92 μmol mL^−1^ (*x* = 0.2).

*Preparation of CCMs with a neat DEGDMA core: crosslinking of the R_0_-(SS^−^Na^+^)_140_-b-(St_1-y_-co-DPPS_y_)_300_-SC(S)SnPr amphiphilic block copolymers*. To the total volume of each of the R_0_-(SS^−^Na^+^)_140_-*b*-(St_1-*y*_-*co*-DPPS*_y_*)_300_-SC(S)S*n*Pr latexes obtained as described above (0.04 mmol of polymer) were successively added H_2_O (1 mL), DEGDMA (0.134 mL, 144.4 mg, 0.6 mmol; 15 equiv. per chain) and the degassed ACPA/NaHCO_3_ stock solution (0.11 mL, 2.2 mg ACPA, 7.96 µmol). The resulting reaction mixtures were stirred at 80 °C for 3 h resulting in an essentially quantitative monomer consumption (<1% of residual DEGDMA by ^1^H NMR monitoring in DMSO-*d*_6_) to yield the CCMs R_0_-(SS^−^Na^+^)_140_-*b*-(St_1-y_-*co*-DPPS*_y_*)_300_-*b*-DEGDMA_15_-SC(S)S*n*Pr (y = 0.05, 0.1, 0.2, 0.25). Polymer content: 18.63% (y = 0.05), 14.26% (y = 0.1), 15.22% (y = 0.2), 15.61% (y = 0.25); [TPP] = 43.71 μmol mL^−1^ (*x* = 0.05), 64.22 μmol mL^−1^ (x = 0.1), 129.39 μmol mL^−1^ (*x* = 0.2), 161.38 μmol mL^−1^ (*x* = 0.25).

*Preparation of NGs*. To a Schlenk tube containing an aqueous solution of R_0_-(SS^−^Na^+^)_140_-SC(S)S*n*Pr aqueous solution (10 mL, 0.04 mmol of polymer, corresponding to 5.55 mmol of SS^−^Na^+^ units), were subsequently added H_2_O (6 mL), 1,3,5-trioxane (ca. 42.1 mg), degassed styrene (1.31 mL, 1.18 g, 11.4 mmol), DPPS (0.173 g, 0.6 mmol) and DEGDMA (0.13 mL, 144.5 mg, 0.6 mmol) and the degassed ACPA/NaHCO_3_ stock solution (0.11 mL, 2.2 mg ACPA, 7.96 µmol). The reaction mixture was stirred at 80 °C for 5 h to yield the NGs R_0_-(SS^−^Na^+^)_140_-*b*-(St_285_-*co*-DPPS_15_-*co*-DEGDMA_15_)-SC(S)S*n*Pr. The latex has a polymer content of 14.6% *w*/*w* ([TPP] = 34.17 μmol mL^−1^).

### 2.3. Synthesis of Phosphine-Functionalized Copolymer Nanoreactors with an Anionic P(SS^−^Na^+^-co-PEOMA) Shell

*Preparation the R_0_-[(SS^−^Na^+^)_0.2_-co-PEOMA_0.8_]_x_-SC(S)SnPr macroRAFT agent (x = 50, 140). A. x = 50.* A portion of the ACPA stock solution (0.5 mL, 10 mg of ACPA, 0.36 mmol), CTPPA (50 mg, 0.18 mmol), SS^−^Na^+^ (372.2 mg, 1.8 mmol; SS^−^Na^+^/CTPPA = 10), PEOMA (6.86 g, 7.22 mmol; PEOMA/CTPPA = 40), ethanol (9 mL) and deionized water (21 mL) were added to a 100 mL Schlenk tube with a magnetic stirrer bar. An internal reference (1,3,5-trioxane, 40.2 mg, 0.47 mmol) was also added for the determination of the monomer conversion as a function of time by ^1^H NMR. The solution was purged for 45 min with argon and then heated to 80 °C over 6 h in a thermostatic oil bath under stirring, leading to quantitative conversion of both monomers. The experimental molar mass (from SEC) for the final polymer is M_n_ = 40,780 g mol^−1^ with *Đ* = 4.95, versus a theoretical molar mass of 40,339 g mol^−1^. The polymer content in the latex is 20.3% *w*/*w*.

*B. x = 140.* This polymerization was carried out under the same conditions as in part A, except for using a lower CTPPA/monomer ratio [CTPPA (50 mg, 0.18 mmol), SS^−^Na^+^ (1.042 g, 5.1 mmol; SS^−^Na^+^/CTPPA = 28), PEOMA (19.21 g, 20.2 mmol; PEOMA/CTPPA = 112), ethanol (12 mL) and deionized water 42 mL). The theoretical molar mass is 112,450 g mol^−1^. The polymer content in the latex is 26.33% *w*/*w*.

*Preparation of latexes of the R_0_-[(SS^−^Na^+^)_0.2_-co-PEOMA_0.8_]_x_-b-St_50_-SC(S)SnPr amphiphilic diblock copolymers*. *A. x = 50.* To the 5 mL pale-yellow latex of the R_0_-[(SS^−^Na^+^)_0.2_-*co*-PEOMA_0.8_]_50_-SC(S)S*n*Pr macroRAFT agent (0.03 mmol of polymer, 7 mL of water) in a Schlenk tube under Ar were added degassed styrene (0.17 mL, 0.15 g, 1.48 mmol; 50 equiv. per chain) and ACPA stock solution (0.1 mL, 2 mg of ACPA, 7.13 μmol). The solution was purged for 45 min with argon and then heated to 80 °C during 5 h in a thermostatic oil bath under stirring, leading to quantitative conversion of styrene. The theoretical molar mass of 45,544 g mol^−1^. The polymer content in the latex is 10.3% *w*/*w*.

*B. x = 140.* This polymerization was carried out under the same conditions as in part A. To the 5 mL pale-yellow latex of the R_0_-[(SS^−^Na^+^)_0.2_-*co*-PEOMA_0.8_]_140_-SC(S)S*n*Pr macroRAFT agent (0.015 mmol of polymer, 5 mL of water) in a Schlenk tube under Ar were added degassed styrene (0.09 mL, 0.078 g, 0.75 mmol; 50 equiv. per chain) and ACPA stock solution (0.05 mL, 1 mg of ACPA, 3.57 μmol). The theoretical molar mass of 117,673 g mol^−1^. The polymer content in the latex is 15.26% *w*/*w*.

*Preparation of latexes of the R_0_-[(SS^−^Na^+^)_0.2_-co-PEOMA_0.8_]_x_-b-(S_0.9_-co-DPPS_0.1_)_300_-SC(S)SnPr amphiphilic copolymers (x = 50, 140). A. x = 50.* To a Schlenk tube under Ar containing the pale-yellow R_0_-[(SS^−^Na^+^)_0.2_-*co*-PEOMA_0.8_]_50_-SC(S)S*n*Pr solution obtained as described above (10 mL, 0.06 mmol of polymer) was added degassed water (16 mL), degassed styrene (1.83 mL, 1.66 g, 15.9 mmol; 270 equiv. per chain) and DPPS (0.51 g, 1.8 mmol; 30 equiv. per chain). A portion of the degassed ACPA/NaHCO_3_ stock solution (0.165 mL, 3.3 mg ACPA, 11.77 µmol) was then added and the resulting reaction mixture was stirred at 80 °C for 5 h, resulting in complete monomer consumption and yielding a white opalescent stable dispersion. The resulting polymer has a theoretical molar mass of 77,117 g mol^−1^. The weight percent polymer in the latex is 15.2% and the phosphine concentration in the latex is 63.04 µmol mL^−1^.

*B. x = 140.* This polymerization was carried out under the same conditions as in part A, except for using the R_0_-[(SS^−^Na^+^)_0.2_-*co*-PEOMA_0.8_]_140_-SC(S)S*n*Pr solution (10 mL, 0.03 mmol of polymer), degassed styrene (0.93 mL, 0.84 g, 15.9 mmol; 270 equiv. per chain), DPPS (0.26 g, 0.9 mmol; 30 equiv. per chain) and degassed ACPA/NaHCO_3_ stock solution (0.1 mL, 2 mg ACPA, 7.14 µmol). Complete monomer consumption led to a stable opalescent dispersion. The resulting polymer has a theoretical molar mass of 149,210 g mol^−1^. The weight percent polymer in the latex is 18.6% and the phosphine concentration in the latex is 42.57 µmol mL^−1^.

*Cross-linking of the P(SS^−^Na^+^-co-PEOMA)-b-P(St-co-DPPS) amphiphilic block copolymers by a DEGDMA-styrene comonomer mixture. Preparation of R_0_-[(SS^−^Na^+^)_0.2_-co-PEOMA_0.8_]_x_-b-(St_0.9_-co-DPPS_0.1_)_300_-b-(S_0.9_-co-DEGDMA_0.1_)_150_-SC(S)SnPr (x = 50, 140). A. x = 50.* To the total volume of the R_0_-[(SS^−^Na^+^)_0.2_-*co*-PEOMA_0.8_]_50_-*b*-(St_0.9_-*co*-DPPS_0.1_)_300_-SC(S)S*n*Pr latex (0.06 mmol of polymer chains) obtained as described above were successively added degassed H_2_O (34 mL), DEGDMA (0.198 mL, 0.214 g, 0.88 mmol; 15 equiv. per chain), styrene (0.91 mL, 0.83 g, 7.94 mmol; 135 equiv. per chain) and the degassed ACPA/NaHCO_3_ stock solution (0.165 mL, 3.3 mg ACPA, 11.8 µmol). The resulting reaction mixture was stirred at 80 °C for 5 h resulting in complete consumption of both monomers (^1^H NMR monitoring in DMSO-*d*_6_). The weight percent polymer in the latex and the phosphine concentration in the latex are 7.27% and 27.89 µmol mL^−1^.

*B. x = 140.* This polymerization was carried out under the same conditions as in part A, except for using the R_0_-[(SS^−^Na^+^)_0.2_-*co*-PEOMA_0.8_]_140_-SC(S)S*n*Pr solution (0.03 mmol of polymer), DEGDMA (0.1 mL, 0.108 g; 0.04 mmol, 15 equiv. per chain), styrene (0.46 mL, 0.42 g, 4.03 mmol; 135 equiv. per chain) and the degassed ACPA/NaHCO_3_ stock solution (0.08 mL, 1.7 mg ACPA, 5.97 µmol). The resulting polymer has a theoretical molar mass of 149,210 g mol^−1^. The weight percent polymer in the latex is 20.3% and the phosphine concentration in the latex is 41.3 µmol mL^−1^.

*Crosslinking of the P(SS^−^Na^+^-co-PEOMA)-b-P(St-co-DPPS) amphiphilic block copolymers by neat DEGDMA. Preparation of R_0_-[(SS^−^Na^+^)_0.2_-co-PEOMA_0.8_]_50_-b-(St_0.9_-co-DPPS_0.1_)_300_-b-DEGDMA_90_-SC(S)SnPr.* To the total volume of the R_0_-[(SS^−^Na^+^)_0.2_-*co*-PEOMA_0.8_]_50_-*b*-(St_0.9_-*co*-DPPS_0.1_)_300_-SC(S)S*n*Pr latex (0.06 mmol of polymer chains) obtained as described above were successively added degassed H_2_O (34 mL), DEGDMA (1.184 mL, 1.281 g, 5.295 mmol; 90 equiv. per chain) and the degassed ACPA/NaHCO_3_ stock solution (0.165 mL, 3.3 mg ACPA, 12.0 µmol). The resulting reaction mixture was stirred at 80 °C for 5 h resulting in complete monomer consumption (^1^H NMR monitoring in DMSO-*d*_6_). The weight percent polymer in the latex and the phosphine concentration in the latex are 8.87% and 27.862 µmol mL^−1^.

*Preparation of NGs*. *A. x = 50.* To a Schlenk tube containing an aqueous solution of R_0_-[(SS^−^Na^+^)_0.2_-*co*-PEOMA_0.8_]_50_-*b*-St_50_-SC(S)S*n*Pr aqueous solution (0.03 mmol of polymer), were subsequently added H_2_O (15 mL), degassed styrene (1.45 mL, 1.31 g, 12.6 mmol), DPPS (0.256 g, 0.89 mmol) and DEGDMA (0.1 mL, 108.2 mg, 0.45 mmol) and the degassed ACPA/NaHCO_3_ stock solution (0.083 mL, 1.7 mg ACPA, 5.92 µmol). The reaction mixture was stirred at 80 °C for 7 h to yield the NG R_0_-[(SS^−^Na^+^)_0.2_-*co*-PEOMA_0.8_]_50_-*b*-St_50_-*b*-(St_425_-*co*-DPPS_30_-*co*-DEGDMA_15_)-SC(S)S*n*Pr. The latex has a polymer content of 10.1% *w*/*w* ([TPP] = 30.78 μmol mL^−1^).

*B. x = 140.* This polymerization was carried out under the same conditions as in part A, except for using the R_0_-[(SS^−^Na^+^)_0.2_-*co*-PEOMA_0.8_]_140_-*b*-St_50_-SC(S)S*n*Pr solution (0.015 mmol of polymer), 5 mL H_2_O, DPPS (0.128 g, 0.44 mmol; 30 equiv. per chain), DEGDMA (0.05 mL, 0.054 g, 0.22 mmol; 15 equiv. per chain), styrene (0.73 mL, 0.66 g, 6.34 mmol; 425 equiv. per chain) and the degassed ACPA/NaHCO_3_ stock solution (0.0418 mL, 0.8 mg ACPA, 2.98 µmol). The weight percent polymer in the latex is 14.9% and the phosphine concentration in the latex is 27.8 µmol mL^−1^.

### 2.4. General Procedure for Rh Complexation to the Phosphine Ligand within CCM or NG Core

In a Schlenk tube was added 1 mL of the polymer latex and 3 mL of H_2_O. Toluene (3 mL) was added and the mixture was stirred for 5 min, resulting in the CCM particle core swelling. Then a separately prepared solution of the desired amount of [RhCl(COD)]_2_, depending on the target P/Rh ratio, in toluene (1 mL) was added to the latex and the mixture was vigorously stirred at room temperature, stopping the stirring at regular intervals (decantation was rapid, <5 min) to assess the progress of the reaction.

### 2.5. General Procedure for the Catalyzed Hydrogenation

In a vial containing a magnetic stirrer was added 0.4 mL of the Rh-charged latex (CCM 10% or NG 10%), prepared as described in the previous sections. The desired amount of styrene was layered on top of the latex. For all experiments, irrespective of the substrate/Rh ratio, decane (internal standard) was then added to the organic layer (substrate/decane molar ratio ca. 4). The vial was then placed inside an autoclave, which was subsequently charged with dihydrogen (20 bar), placed in a thermostatic oil bath, and stirred at 1200 rpm. At the set reaction time, the stirring was stopped, the autoclave was vented and the vial was taken out under argon. The latex decantation was rapid (<5 min). After phase separation, the latex was extracted with toluene (3 × 0.3 mL). The combined organic phases were used for the GC analysis. For the recycling experiments, a fresh substrate solution (same amounts as in the initial run) was added to the same vial, followed by reaction and product separation according to the same protocol.

## 3. Results

### 3.1. TPP-Functionalized CCMs and NGs with Hydrophilic P(SS^−^Na^+^) Homopolymer Blocks

The synthesis of the new polymers followed the same convergent strategy previously used for the construction of CCMs and NGs with a neutral [21,25] or polycationic [32] outer shell, as already outlined in the Introduction, except for the nature of the hydrophilic blocks assembled in the first step, as schematically shown in Figure 2. The final latexes contain up to ca. 27% of polymer content in mass and have low viscosity, as expected for stable suspensions of spherical micelles. The suspensions are stable, with no evidence of coagulation over several months, but must be stored under an inert atmosphere to avoid the slow aerial oxidation of the phosphine ligand to phosphine oxide. For convenience, all new polymers described in this section, as well as references to their characterization data, are summarized in the Appendix A. As additional help to the reader, the polymer formulas in the manuscript text and Appendix A are written using the color coding of Figure 2. The α and ω chain ends of the macromolecule are provided by the RAFT chain transfer agent 4-cyano-4-thiothiopropylsulfanyl pentanoic acid (CTPPA) (R_0_ = −C(CH_3_) (CN) CH_2_CH_2_COOH). We have already optimized and recently published [34] the synthesis of unimolecular CCMs that are fully equivalent to those described here, except for the absence of ligand-functionalized monomer. As a short summary, the aqueous dispersions of the micelles formed by the intermediate R_0_-(SS^−^Na^+^)*_x_*-*b*-St*_y_*-SC(S)S*n*Pr diblock copolymer were found unstable, featuring equilibria with single chains and large agglomerates, especially for polymers with a short PSt block. However, the optimized polymer compositions (*x* = 140, *y* > 300) led to well-defined CCMs with narrow size distributions after the final cross-linking step. For this reason, the synthesis of the ligand-functionalized CCMs was pursued by fixing the degrees of polymerization to 140 for the outer hydrophilic P(SS^−^Na^+^) shell and to 300 for the PSt core. The fraction of DPPS (a solid monomer) in the St/DPPS mixture is limited by the DPPS solubility in styrene (ca. 25% molar), because the suspension polymerization in the second step of the CCM synthesis requires the presence of only two liquid phases in order to yield full monomer conversion and produce well-defined micelles.

Given previously encountered difficulties (precipitation of DPPS as a result of the faster incorporation of styrene, especially when using the higher DPPS fractions), a few CCMs were also initially developed by first extending the R_0_-(SS^−^Na^+^)_140__−_SC(S)S*n*Pr macromolecules with a short PSt block (50 monomer units), yielding an amphiphilic diblock copolymer, which self-assembles. Further chain extension of the R_0_-(SS^−^Na^+^)_140_-*b-*St_50_-SC(S)S*n*Pr macroRAFT agent with the St/DPPS mixture then starts off directly as an emulsion polymerization, removing any potential DPPS precipitation issues and ensuring full incorporation of the DPPS monomer in the CCM core. Indeed, the ^1^H NMR monitoring indicates full consumption of all monomers (Appendix A). However, this precaution was later found superfluous, as essentially complete DPPS incorporation (see Appendix A) without precipitation also took place upon direct extension of the hydrosoluble R_0_-(SS^−^Na^+^)_140_-SC(S)S*n*Pr macroRAFT agent, even for a 25% molar DPPS fraction. The DLS and TEM characterization of most of the resulting dispersions is collected in Appendix A. The DLS measurements indicate the presence of micelles with average diameter in the 50–80 nm range, although distributions of larger agglomerates are also observed for the unfiltered dispersions. This behavior is quite similar to that observed for the equivalent diblock copolymers with a neat PSt block [34]. The NMR spectra of the self-assembled di/triblock copolymers in DMSO-*d*_6_ (Appendix A) only reveal the solvated P(SS^−^Na^+^) block resonances, because DMSO is a bad solvent for polystyrene and the (St_1-*y*_-*co*-DPPS*_y_*)_300_
or St_50_-*b*-(St_1-*y*_-*co*-DPPS*_y_*)_300_
cores do not have sufficient mobility. All resonances, however, became visible in D_2_O after swelling the cores with CDCl_3_ (see representative examples in Appendix A).

The final cross-linking step was carried out using either a 10:90 DEGDMA/St comonomer mixture or neat DEGDMA, using in all cases 15 DEGDMA units per macromolecular chain. As detailed in our recent contribution [34], using this amount of neat DEGDMA is sufficient to ensure quantitative cross-linking of all diblock copolymers without macrogelation and generate single spherical CCMs with a narrow size distribution, as is also the case for the equivalent CCMs with a polycationic P(4VPMe^+^I^−^) shell [31,32]. Conversely, CCMs with a neutral P(MAA-*co-*PEOMA) shell could be crosslinked without macrogelation only when using a styrene-rich DEGDMA/St comonomer mixture [21]. All triblock copolymers (i.e., containing an intermediate St_50_ homopolymer block) were crosslinked with the 10:90 DEGDMA/St comonomer mixture with quantitative incorporation of DEGDMA, only occasionally leaving a small residual amount of unreacted styrene (Appendix A), to yield CCMs of composition R_0_-(SS^−^Na^+^)_140_-*b-*St_50_-*b-*(St_1-*y*_-*co-*DPPS*_y_*)_300_-*b-*(St_0.9_-*co*-DEGDMA_0.1_)_150_-SC(S)S*n*Pr (*y* = 0.05, 0.10, 0.20). The DLS and TEM characterization of these polymers is summarized in Figure 3. The presence of a minor amount of aggregates is indicated by the DLS traces of the unfiltered dispersions (see Appendix A for *y* = 0.05, Appendix A for *y* = 0.1 and Appendix A for *y* = 0.2), but the filtered samples reveal only monomodal distributions with diameters close to 100 nm and the TEM images, confirm their spherical morphology. Interestingly, after swelling with toluene or chloroform, the average diameters of the particles do not significantly increase, contrary to what was observed in the similar TPP-free CCM particles [34]. However, swelling is clearly evidenced by the ^31^P NMR characterization: the TPP resonance at δ ca. −6.5 becomes visible only after swelling the cores with CDCl_3_ (Appendix A). The probable reason for the *D*_z_ decrease upon addition of swelling solvents is a disaggregation action, reducing the impact of larger-size agglomerates and masking the size increase of the swollen cores.

All diblock copolymers (without an intermediate St_50_ homopolymer block), on the other hand, were only cross-linked with neat DEGDMA, once again with quantitative or nearly quantitative monomer incorporation (Appendix A) to yield CCMs of composition R_0_-(SS^−^Na^+^)_140_-*b*-(St_1-*y*_-*co-*DPPS*_y_*)_300_-*b-*DEGDMA_15_-SC(S)S*n*Pr (*y* = 0.05, 0.10, 0.20, 0.25). The NMR characterization in D_2_O/CDCl_3_ is available in Appendix A. The DLS results (Figure 4) are very similar to those of the CCMs cross-linked with the St/DEGDMA comonomer mixture, at least for the particles with lower DPPS fraction (*y* = 0.05, 0.10). For those with higher DPPS fraction (*y* = 0.20, 0.25), the unswollen latexes show a greater contribution of larger size distributions, probably CCM agglomerates, which are particularly visible in the DLS of the unfiltered latexes (Appendix A for *y* = 0.05, Appendix A for *y* = 0.10, Appendix A for *y* = 0.20 and Appendix A for *y* = 0.25), but these are almost completely absent in the DLS of the swollen latexes, while the TEM images show the predominance of individual spherical particles (Figure 4). This is further evidence of the disaggregation action of the swelling solvent, which causes once again a reduction of the *D*_z_ values. 

A compatibilizing solvent, able to solvate both the P(SS^−^Na^+^) shell and the P(St-*co-*DPPS) core and to break down the micelles of the diblock copolymer intermediate into single chains [35], was searched for the purpose of probing the cross-linking completeness (absence of residual uncrosslinked diblock arms in the CCM product by DLS analysis). For the very closely related TPP-free polymers, this analysis was successfully carried out in THF-water or DMF-water mixtures with >40% water: the DLS of the P(SS^−^Na^+^)-*b*-PSt diblock chains revealed small objects (d < 10 nm), while that of the corresponding P(SS^−^Na^+^)-*b*-PSt-*b*-PDEGDMA and P(SS^−^Na^+^)-*b*-PSt-*b*-P(St-*co*-DEGDMA) cross-linked products proved the absence of a small diameter population, hence demonstrating the quantitative linking of all diblock arms [34]. The introduction of the DPPS comonomer in the hydrophobic block, on the other hand, did not allow finding suitable conditions for a complete solvation of the uncrosslinked micelles in the form of single chains, probably because the incorporated DDPS renders the PSt block even more hydrophobic and perturbs the single chain—micelle equilibrium in favor of the latter. On the basis of the results presented in our previous investigation for the TPP-free CCM particles and the equivalence of the block composition (except for the DPPS incorporation) and polymerization conditions, we assume that the TPP-functionalized CCM are also characterized by quantitative cross-linking of all diblock arms.

Unimolecular amphiphilic copolymers with an NG architecture and an outer P(SS^−^Na^+^) shell have not been previously described, to the best of our knowledge. The straightforward extension of the strategy that leads to NG nanoreactors with a neutral or polycationic hydrophilic shell to the P(SS^−^Na^+^)-based macroRAFT agents (Figure 2) led to complete monomer conversion and stable latexes of well-defined NG particles with narrow size distributions. Particles with two different compositions were obtained, one starting from R_0_-(SS^−^Na^+^)_140_-SC(S)S*n*Pr and the second one starting from R_0_-(SS^−^Na^+^)_140_-*b*-St_50_-SC(S)S*n*Pr. The two products also differ by the amounts of St and DPPS comonomers. Characterization data available in the Supporting Information attest the quantitative conversion of all comonomers (by ^1^H NMR, Appendix A), except for a small amount of residual styrene in one case, and the quality of the resulting CCM and NG particles (DLS of unswollen *vs.* toluene- and chloroform-swollen latexes, Appendix A for *x*, *y*, *z* = 0,285,15 and Appendix A for *x*,*y*,*z* = 50, 425, 30). The DLS results of the filtered latexes are also shown in Figure 5, together with TEM images. In both cases, the size distribution in the number mode has an average diameter under 100 nm, though the presence of larger aggregates is shown by the volume and intensity distributions and by the high PDI values, particularly for the sample in Figure 5b. As already shown above for the CCM particles, treatment with swelling solvents leads to lower *D*_z_ and PDI values, which is attributed to reduced aggregation.

### 3.2. Treatment with [RhCl(COD)]_2_ and Model Studies

Following an identical protocol to that previously implemented to load the 1st and 2nd generation CCM/NG nanoreactors with the [RhCl(COD)]_2_ precatalyst [21,25,26,32], the latexes of the anionic-shell polymers described above were first core-swollen by toluene. However, the subsequent addition of the [RhCl(COD)]_2_ toluene solution led to the immediate polymer coagulation, leaving a transparent and colorless supernatant liquid (see Appendix A). Since no such behavior was previously observed, neither for the neutral P(MAA-*co*-PEOMA)-based 1st generation nanoreactors nor for the cationic P(4VPMe^+^I^−^)-based 2nd generation ones, it seems likely that the rhodium complex interacts with the particle P(SS^−^Na^+^) shell, leading to particle aggregation and macrogelation. In order to find supporting evidence and to learn more about this phenomenon, a few tests were carried out with models of the CCM particle surface, starting with two molecular salts (sodium styrenesulfonate and *p*-tolylsulfonate, NaO_3_SC_6_H_4_-4-R with R = vinyl and methyl, respectively), both in the absence and presence of PPh_3_. We have not been able to find previously published reports on the coordination chemistry of [RhCl(COD)]_2_ and [RhCl(COD)(PPh_3_)] with sulfonate salts. These rhodium complexes are frequently used as precursors to generate cationic precatalysts with the [Rh(COD)L_2_]^+^X^−^ stoichiometry (L = monodentate or L_2_ = bidentate ligand; X = non-coordinating anions, e.g., BF_4_^−^, PF_6_^−^, B[3,5-C_6_H_3_(CF_3_)_2_]_4_^−^, etc.), which also include several trifluoromethylsulfonate (triflate, OTf^−^) salts, e.g., [Rh(COD)(R-DuPHOS)]^+^OTf^−^ (R = Me, Et, *i*Pr) [36,37], but examples with other alkyl or arylsulfonates anions do not apparently exist. On the other hand, the coordinating properties of arylsulfonates are well established for phosphine-sulfonate chelating ligands in palladium polymerization catalysts [38].

The toluene solutions of complexes [RhCl(COD)]_2_ (yellow) and [RhCl(COD)(PPh_3_)] (red, produced in situ from [RhCl(COD)]_2_ and PPh_3_ in a 1:2 ratio) were layered on top of an aqueous solution containing an excess of the sulfonate salt, followed by vigorous stirring at room temperature for 3 h. The Rh/salt ratio (ca. 1:5) was adjusted to approximately reproduce the ratio used in the polymer loading experiments. When using the styrene sulfonate salt, the aqueous phase remained essentially colorless after decantation (Figure 6), while the aspect of the supernatant organic later was unchanged.

When using the *p*-tolylsulfonate salt, on the other hand, the aqueous phase became pale yellow (more so in the absence of PPh_3_ than in its presence). These results reveal a certain degree of interaction between the Rh center and ArSO_3_^−^ (Figure 1). In the absence of PPh_3_, the putative anionic chloro-sulfonate [RhCl(COD)(O_3_SAr)]^−^ complex or a bis-sulfonate [Rh(COD)(O_3_SAr)_2_]^−^ product of ligand exchange (reaction (1) in Figure 1) is partially transferred to the aqueous phase. In the presence of PPh_3_, on the other hand, sulfonate addition to make a 5-coordinate [RhCl(COD)(PPh_3_)(O_3_SAr)]^−^ (reaction (2) in Figure 1) is less favored, while a possible chloride/sulfonate exchange product, [Rh(COD)(PPh_3_)(O_3_SAr)], being uncharged, might prefer to remain in the toluene phase. A weak interaction must nevertheless be present, because the aqueous solution is not completely colorless. The more extensive interaction of *p*-tolylsulfonate relative to SS^−^Na^+^ may result from either a stronger coordinating ability or a greater hydrophilicity of the resulting rhodium complexes.

Evidence in favor of a Rh-sulfonate interaction was also provided by a ^1^H NMR investigation using DMSO-*d*_6_/D_2_O as a compatibilizing solvent mixture. Upon mixing [RhCl(COD)]_2_ and NaO_3_SC_6_H_4_-4-CH_3_ (Rh/sulfonate = ca. 1:5), a slight upfield shift was observed for all three COD resonances (from 4.45, 2.42 and 1.99 ppm to 4.41, 2.36 and 1.93 ppm, respectively) and for one of the aromatic sulfonate resonances (7.55–7.52 ppm), while the other resonances of the sulfonate *p*-tolyl group (aromatic at 7.15 ppm and CH_3_ at 2.27 ppm) remained essentially unshifted, see Appendix A. On the other hand, the [RhCl(COD)(PPh_3_)]/NaO_3_SC_6_H_4_-4-CH_3_ interaction (Rh/sulfonate = ca. 1:5) did not reveal any significant shift for the COD ^1^H NMR resonances, but a slight upfield shift for the sulfonate aromatic resonances was again observed, see Appendix A.

Additional experiments were carried out by layering the toluene [RhCl(COD)]_2_ and [RhCl(COD)(PPh_3_)] solutions on top of an aqueous solution of the P(SS^−^Na^+^) macroRAFT chains (degree of polymerization = 140), using again the same Rh/sulfonate ratio. In this case, the Rh species is nearly completely transferred to the aqueous phase in the absence of PPh_3_, to yield a transparent red solution, see Figure 7. The more extensive extraction of [RhCl(COD)]_2_ into the aqueous phase by the macroRAFT chain is probably favored by the formation of a bis-sulfonate complex through the action of the chelate effect, which is most likely (but perhaps not exclusively) implemented through the use of two adjacent monomer units in the chain. Therefore, a sulfonate function appears able to replace the chloride ligand (reaction (3) in Figure 2). This result rationalizes the polymer coagulation observed when [RhCl(COD)]_2_ is added to the colloidal dispersion of the CCM or NG polymers: complexation to form a species with a coordination environment such as the product of reaction (3) in Figure 2 is also possible when the two sulfonate ligands are provided by two different polymer particle shells.

Treatment of the aqueous macroRAFT solution with a toluene solution of [RhCl(COD)(PPh_3_)], on the other, led to a turbid toluene phase containing a large amount of orange coagulated polymer, and a transparent pale yellow aqueous phase, see Figure 7. This confirms the ability of the sulfonate groups to bind the Rh center even in the presence of PPh_3_. The greater interaction of the macroRAFT agent relative to the molecular salt suggests replacement of the chloride ligand and formation of a 5-coordinate anionic complex, once again favored by the action of the chelate effect (reaction (4) in Figure 2). The presence of the tightly bonded hydrophobic phosphine ligand renders the product insoluble in water, while the presence of charged sulfonate functions and sodium counter-ions make it also insoluble in toluene, hence leading to precipitation.

^1^H NMR investigations to probe the presence and the effect of the Rh-sulfonate interaction were again carried out in DMSO-*d*_6_/D_2_O, yielding similar results as the experiments with the molecular salts. In the absence of PPh_3_ (Appendix A), the three COD resonances slightly shifted upfield (from 4.46, 2.42 and 1.99 ppm to 4.42, 2.36 and 1.93 ppm, respectively). In this case, the broader aryl and backbone aliphatic resonances of the macroRAFT chain did not show any significant shift. For the experiment with [RhCl(COD)(PPh_3_)], Appendix A, which yielded a homogeneous solution, the change was less perceptible, with possibly a slight upfield shift of the broad resonances centered at ca. 2.33 and 1.95 ppm.

### 3.3. Synthesis of CCMs and NGs with a P(SS^−^Na^+^-co-PEOMA)-Based Shell

Given that the P(SS^−^Na^+^) homopolymer chains not only block the [RhCl(COD)]_2_ migration through the shell, but even induce polymer coagulation, our shell engineering efforts continued with a decrease of the anionic monomer density, via dilution with a neutral monomer. The working hypothesis was that an increased average distance between the SS^−^Na^+^ monomers may weaken the Rh-shell interactions by removal of the chelate effect. Consequently, polymer coagulation may be disfavored, and the Rh complexes may be able to migrate through the shell to eventually encounter the much stronger-binding TPP ligands in the core. As a diluting monomer, PEOMA was the natural choice on the basis of the previous incorporation of this monomer in the 1^st^-generation neutral-shell particles [21,22,23,24,25,26,27,28]. The SS^−^Na^+^ fraction in the [(SS^−^Na^+^)*_x_*-*co*-PEOMA_1-*x*_] shell blocks must be small enough to reduce the shell ability to capture the Rh complex, but sufficient to keep a significant shell-shell repulsion to avoid interpenetration and to maintain high-temperature hydrophilicity. All polymers developed in this contribution were made with the fixed molar fraction *x* = 0.2 (i.e., 20% of SS^−^Na^+^ monomers in the hydrophilic block). All these polymers are summarized in Appendix A, which also serves as a reference for the characterization data.

The synthesis followed the same general procedure described above, except for the use of a 20:80 SS^−^Na^+^/PEOMA comonomer mixture in the first step (Figure 8), yielding R_0_-[(SS^−^Na^+^)_0.2_-*co*-PEOMA_0.8_]*_x_*-SC(S)S*n*Pr. Two different degrees of polymerizations (*x* = 50, 140) were targeted to probe the effect of the hydrophilic shell thickness on the latex stability. Indeed, while the best charged-shell particles were formed with a high degrees of polymerization (140) for the P(SS^−^Na^+^) [34] and P(4VPMe^+^I^−^) [31] homopolymer blocks, those with a P(MAA-*co*-PEOMA)-based neutral shell [21] only required a low degree of polymerization (30).

^1^H NMR monitoring of the polymerization indicated essentially complete incorporation of both monomers (only traces of residual PEOMA were visible, see Figure 9 for *x* = 50 and Appendix A for *x* = 140). Further analysis of the final latex ^1^H NMR spectrum in DMSO-*d*_6_ shows interesting features. Whereas the PEO resonances (OC*H*_2_C*H*_2_O at 3.48 ppm and OC*H*_3_ at 3.21 ppm in the monomer) remain essentially unchanged in terms of their position and sharpness, the resonances of the methacrylate CH_3_ protons at 1.84 ppm and of the SS^−^Na^+^ aryl protons at 7.4 and 6.9 ppm were extensively broadened after incorporation into the polymer chains. This demonstrates that the polymer backbone is much less solvated, and thus less mobile in solution relative to the PEO side chains. These features will be of interest in comparison with those shown below for the CCM and NG polymers. Another point to note is that the water and ethanol OH protons give independent resonances in DMSO-*d*_6_, indicating slow chemical exchange that can be attributed to the strong H-bonding with the proton-accepting solvent. The chemical shift of these two resonances changes from the initial to the final solution.

A more detailed monitoring of the polymerization with *x* = 50 revealed a much faster consumption of SS^−^Na^+^. When the charged monomer was completely consumed, only ca. 60% of the PEOMA had been incorporated (Appendix A). The SEC monitoring (Appendix A) showed that the polymer molar mass increased more or less linearly with respect to the overall monomer conversion, although with quite high polymer dispersity (>2). Therefore, the resulting water-soluble chains have a marked gradient topology with a greater SS^−^Na^+^ fraction at the α-end (R_0_) and an essentially pure P(PEOMA) homopolymer sequence at the ω-end (trithiocarbonate). The DLS analysis of the R_0_-[(SS^−^Na^+^)_0.2_-*co*-PEOMA_0.8_]*_x_*-SC(S)S*n*Pr (*x* = 50, 140) solutions, after extensive dilution with water, shows a dominant distribution of single chains (average diameter ca. 1 nm), plus a minor population of larger aggregates (*d* = 20–30 nm) that is clearly visible only in the intensity mode (Appendix A). The TEM images of these polymers, also shown in Appendix A, show only ill-defined aggregates with no evidence of regular self-organization. The analogous macroRAFT agents with the same degrees of polymerization made of SS^−^Na^+^ homopolymer chains displayed very similar average size in solution and morphology as a dry polymer [34]. In combination with the NMR evidence discussed above, the organization of these objects in the DMSO solution probably consist of a rather contracted backbone in the particle core, particularly for the SS^−^Na^+^-rich end of the gradient copolymer, surrounded by the better solvated PEO chains. This statement seems further supported by the ^1^H NMR spectrum of these two polymers in D_2_O (Appendix A), in which the resonances of the PEO chains (backbone methylene and chain-end methyl resonances at δ 3.63 and 3.31 ppm, respectively) are visible as narrow peaks (w_1/2_ ~ 5 Hz), as expected for well-solvated protons with short correlation times, whereas those of the styrenesulfonate aromatic protons (at δ 7.56 and 7.11 ppm, w_1/2_ ~ 60 and 80 Hz) and main-chain aliphatic protons, including the methacrylate Me protons (in the δ 2–0.5 ppm region), are much broader. The additional broad resonance observed at δ 4.1 ppm (w_1/2_ ~ 60 Hz), slightly downfield-shifted from the strong PEO methylene resonance, is assigned to the first PEO CH_2_ group bonded to the methacrylate ester function; its greater linewidth results from its proximity to the less mobile polymer chain.

These macroRAFT agents were chain-extended by either a short PSt homopolymer block (50 monomer units per chain) or a longer P(St-*co*-DPPS) copolymer block (300 monomers per chain, with the DPPS fraction fixed at 10%). Stable latexes were obtained in all cases. For the R_0_-[(SS^−^Na^+^)_0.2_-*co*-PEOMA_0.8_]*_x_*-*b-*St_50_-SC(S)S*n*Pr (*x* = 50, 140) diblock products, the DLS and TEM characterization is shown in Appendix A. The *x* = 50 sample shows a dominant population with ca. 5 nm average diameter and a second population around 50 nm (more evident in the intensity mode, Appendix A), suggesting the formation of small aggregates and micelles. The presence of spherical micelles is confirmed by TEM. For the *x* = 140 sample (Appendix A), the DLS shape is similar but the lower-size population has a smaller average diameter (ca. 1 nm), more consistent with the presence of single chains, while the TEM analysis reveals once again the presence of spherical micelles. Thus, single-chain/micelle or smaller/larger micelle equilibria appear to be present in water. For the R_0_-[(SS^−^Na^+^)_0.2_-*co*-PEOMA_0.8_]*_x_*-*b-*(St_0.9_-*co-*DPPS_0.1_)_300_-SC(S)S*n*Pr (*x* = 50, 140) products (^1^H NMR monitoring shown in Appendix A), only the sample with *x* = 50 was investigated by DLS and TEM prior to cross-linking (Appendix A). These analyses revealed a dominant population of small spherical micelles (*d* ca. 8 nm from the DLS) and a minor population of larger objects (*d* ca. 80 nm) that appeared to be vesicles from the TEM image.

The ^1^H NMR spectra of the diblock copolymers in DMSO-*d*_6_ (Appendix A) reveal interesting features: the SS^−^Na^+^ resonances of the P(SS^−^Na^+^-*co*-PEOMA) shell, which are broad but visible in the spectrum of the water-soluble macroRAFT agent at δ 7.4 and 6.8 (Appendix A), are not only broad but also barely observable with much smaller than expected relative intensities when compared with the sharp and intense resonances of the well-solvated PEO. The resonances of the P(St-*co-*DPPS) core are also not observed in DMSO-*d*_6_ but become visible after dispersion in a D_2_O/CDCl_3_ solvent mixture, while those of the shell backbone and SS^−^Na^+^ arene protons remain unobserved. This suggests that the shell backbone remains unsolvated, and is probably located at the interface between the CDCl_3_-swollen core and the outer aqueous solution. This behavior is identical to that previously described for the P(MAA-*co*-PEOMA) backbone of the 1st-generation neutral-shell CCM and NG polymers [26]. It can thus be concluded that the shell PEOMA monomers provide greater stabilization than the SS^−^Na^+^ monomers to the water dispersions of the self-assembled diblock copolymers.

Like for the P(SS^−^Na^+^)-shell CCM particles described in Section 3.1, the final cross-linking step was carried out with either a DEGDMA/styrene comonomer mixture or neat DEGDMA, both leading to stable latexes of spherical particles with a narrow size distribution. Cross-linking with the DEGDMA/styrene mixture gave polymers with average composition R_0_-[(SS^−^Na^+^)_0.2_-*co*-PEOMA_0.8_]*_x_*-*b-*(St_0.9_-*co*-DPPS_0.1_)_300_-*b*-(St_0.9_-*co*-DEGDMA_0.1_)_150_-SC(S)S*n*Pr (*x* = 50, 140), whereas neat DEGDMA was used to obtain particles with average composition R_0_-[(SS^−^Na^+^)_0.2_-*co*-PEOMA_0.8_]_50_-*b*-(St_0.9_-*co*-DPPS_0.1_)_300_-*b*-DEGDMA_90_-SC(S)S*n*Pr. The NMR monitoring and product characterization is available in Appendix A and the DLS and TEM characterization is reported in Figure 10. The DLS of the two samples with the [(SS^−^Na^+^)_0.2_-*co*-PEOMA_0.8_]_50_
hydrophilic block, (a) and (c), reveal the presence of large aggregates in the unfiltered samples, but the size distributions become essentially monomodal after filtration, whereas the polymer with the [(SS^−^Na^+^)_0.2_-*co*-PEOMA_0.8_]_140_
hydrophilic block shows a monomodal distribution also in the unfiltered sample, (b). In all cases, the distribution is peaked around a 20–30 nm diameter in the number mode. The TEM images confirm the spherical morphology of the polymer particles.

Simultaneous chain extension and cross-linking of the R_0_-[(SS^−^Na^+^)_0.2_-*co*-PEOMA_0.8_]*_x_*-*b*-St_50_-SC(S)S*n*Pr (*x* = 50, 140) macroRAFT agents with the St/DPPS/DEGDMA mixture (425:30:15 monomers per chain) led to NG polymers, the NMR and DLS/TEM characterization of which is reported in Appendix A, respectively. The resulting particles have once again spherical topology and the most probable diameter resulting from the DLS distribution in the number mode is significantly smaller for the *x* = 50 sample (ca. 10 nm) than for the *x* = 140 sample (ca. 30 nm), but both are contaminated by larger size objects, clearly visible in the intensity mode.

A change of shell composition from P(SS^−^Na^+^) to P[(SS^−^Na^+^)_0.2_-*co*-PEOMA_0.8_] allowed the micelles of the diblock copolymer intermediate, P[(SS^−^Na^+^)_0.2_-*co*-PEOMA_0.8_]-*b*-P(St-*co-*DPPS), to be broken down into smaller objects by using a THF-water 60:40 mixture, although repeated measurements indicated a metastable situation with oscillation between very small objects (<0.5 nm) and medium-size ones (d ~ 1 nm), together with a small number of larger aggregates (visible only in the intensity distribution), see the DLS of R_0_-[(SS^−^Na^+^)_0.2_-*co*-PEOMA_0.8_]_50_-*b*-(St_0.9_-*co*-DPPS_0.1_)_300_-SC(S)S*n*Pr in Appendix A. The greater hydrophilicity of the P[(SS^−^Na^+^)_0.2_-*co*-PEOMA_0.8_] shell, as suggested by the ^1^H NMR investigation, is probably responsible for a change of organization, tilting the self-organization equilibrium towards single chains in this mixed solvent. The DLS in the same solvent mixture of the corresponding R_0_-[(SS^−^Na^+^)_0.2_-*co*-PEOMA_0.8_]_50_-*b*-(St_0.9_-*co*-DPPS_0.1_)_300_-*b-*(St_0.9_-*co*-DEGDMA_0.1_)_150_-SC(S)S*n*Pr CCM, on the other hand, yielded a stable dispersion of micelles with narrow size distribution (D_z_ = 119 nm, PDI = 0.75) and no visible presence of smaller objects (Appendix A), demonstrating the completeness of the cross-linking reaction.

### 3.4. [RhCl(COD)]_2_ Loading in the CCM and NG with the P(SS^−^Na^+^-co-PEOMA)-Based Shell

Treatment of the toluene-swollen CCMs and NGs with a P(SS^−^Na^+^-*co*-PEOMA) shell with a toluene solution of [RhCl(COD)]_2_ resulted in transfer of the Rh complex to the latex phase without any polymer coagulation, as visually evident from the discoloration of the organic phase and the formation of a pale orange-colored and stable aqueous dispersion. Therefore, the SS^−^Na^+^ dilution strategy has successfully decreased the shell ability to interact with the Rh^I^ center. Coordination of the Rh complex to the core-anchored TPP ligands is demonstrated by the replacement of the uncoordinated TPP ^31^P NMR signal at δ ca. −6 ppm with the characteristic signal [21,25,26,32] of the [RhCl(COD)(TPP@CCM)] complex at δ 29.4 ppm (^2^J_RhP_ = 150 Hz), see Figure 11. As described previously, this signal is visible only when using a P/Rh ratio of 1:1 or smaller, because the presence of free TPP results in signal coalescence and broadening beyond detection at room temperature, because of a rapid degenerative exchange between free and coordinated TPP [21]. The catalytic runs (next section), however, were carried out under the previously optimized conditions, namely with a P/Rh ratio of 4.

### 3.5. Hydrogenation Catalysis

The performance of the Rh-loaded CCM and NG nanoreactors with a P(SS^−^Na^+^-*co*-PEOMA) shell was evaluated in the aqueous biphasic hydrogenation of neat styrene. Previous work with neutral and polycationic-shell nanoreactors have indicated the presence of an initial catalyst activation phase, a limited impact of mass transport kinetics (stirring rate-dependent TOF) [22,25], catalytic activities up to >300 h^−1^ at 25 °C and catalyst leaching down to ca. 0.1 ppm (for the polycationic-shell polymers). Under typical conditions, kinetic experiments carried out with two different nanoreactors confirmed the presence of an activation phase within the first couple of hours, followed by rapid substrate consumption and leading to complete conversion within ca. 10 h (average TOF = ca. 200 h^−1^), see Figure 12. The two polymers used in these experiments differ in the thickness of the hydrophilic shell (DP of 140 or 50) and in the composition of the cross-linking monomers [(St_0.9_-*co*-DEGDMA_0.1_)_150_ vs. DEGDMA_90_], but only the first variation is expected to potentially impact mass transport. The rough equivalence of the results indicates a minor impact of the shell thickness on the mass transport kinetics.

In terms of selectivity, all hydrogenations produced 100% ethylbenzene, without any ring hydrogenation to ethylcyclohexane, in accord with the previously reported performance of CCM/NG-embedded [RhCl(COD)(TPP)] precatalysts [27,32]. This is consistent with the absence of metal reduction to generate metallic Rh nanoparticles (NPs), which is also attested by the absence of color change for the recovered latex after catalysis, even after multiple recycles (*vide infra*). From previous research carried out in our laboratory, we know that Rh NPs can indeed be generated by H_2_ reduction of core-anchored [RhCl(COD)(TPP)], but this only occurs in the absence of styrene, which acts as a π-acidic ligand to stabilize molecular Rh^I^, and is accelerated by heating and by organosoluble bases (e.g., Et_3_N) [39].

Recycling experiments carried out with both the CCM- and NG-embedded catalyst gave puzzling results (Figure 13 and detailed data in Appendix A) when compared with those of the equivalent nanoreactors with the polycationic P(4VPMe^+^I^−^) shell. First of all, the decantation process was not as rapid and clean [32], leaving evident opacity in the organic phase (see Appendix A). The previously investigated polycationic nanoreactors gave increased conversions after recycling, because of the pre-catalyst activation phase. Conversely, the present nanoreactors led to a significant activity decrease after the first run, followed by approximately constant activity in subsequent recycles. The activity drop was more substantial for the two nanoreactors with a thicker outer shell (DP = 140, Figure 13a,c) and smaller for the shorter shell chain nanoreactor (DP = 50, Figure 13b). This phenomenon suggests the possible intervention of the shell sulfonate groups in the modification of the catalyst activity after pre-catalyst activation. During the activation phase, the COD ligand is presumably hydrogenated and removed to yield a less saturated Rh^I^ center, which may find a better stabilization by interacting with the sulfonate groups at the core-shell interface.

In order to validate this hypothesis, control runs were carried out with a variety of model catalysts, see Table 1, using the same conditions and over a constant reaction time of 2.5 h. In comparison with the results given by the CCM (entries 1,2) and NG (entry 3) nanoreactors, all other runs provide higher TONs, a probable consequence of mass transport limitations for the catalytic nanoreactors. The molecular catalyst obtained from [RhCl(COD)]_2_/PPh_3_ is very active (entry 4), but intriguingly this activity dramatically drops when the reaction was carried out in the absence of a water phase (entry 5), even though all needed components are located in the organic phase. This phenomenon (improved reaction rates for reactions run “on water”) has also been highlighted in other cases [40,41]. When using sulfonate-containing additives without phosphine (*p*-toluenesulfonate, entry 6, or the R_0_-[(SS^−^Na^+^)_0.2_-*co*-PEOMA_0.8_]_50_-SC(S)S*n*Pr macroRAFT agent, entry 7), very high activities were again obtained. However, these experiments led to the formation of metallic Rh NPs (see Appendix A), which are notoriously very active hydrogenation catalysts [42,43]. Whenever phosphine ligands were present in the system, under the conditions of these catalytic runs there was no evidence for the formation of Rh NPs. The combination of sulfonate groups and PPh_3_ (entries 8–10) led to significantly reduced activities with respect to the experiment run with PPh_3_ only (entry 4), although higher than for those with the nanoreactors. Therefore, these results validate the hypothesis that the shell sulfonate groups negatively interfere with the catalytic performance of the Rh^I^ active sites.

In addition to the unsatisfactory recycling performance, the ICP-MS analysis of the recovered organic phases revealed substantial catalyst leaching (up to 26.6 ppm of Rh, Appendix A). The catalyst loss must be related to the transfer of the entire catalytic nanoreactors to the organic phase, not to the loss of molecular Rh species, also indicated by the above-mentioned opacity of the recovered organic phases. The DLS analysis of these phases confirmed the presence of the nanoreactors, with dimensions closely comparable to those measured in the initial latex (Appendix A), without any evidence for aggregation. Therefore, in terms of both long-term stability and leaching, these polymers are of lower interest than those with a neutral P(MAA-*co-*PEOMA) shell and especially than those with a polycationic P(4VPMe^+^I^−^) shell.

## 4. Conclusions

Unimolecular amphiphilic core-shell star-block polymers with a triphenylphosphine-functionalized hydrophobic core and an anionic hydrophilic shell based on styrene sulfonate monomers have been synthesized, loaded with [RhCl(COD)]_2_, and applied to the aqueous biphasic hydrogenation of styrene. The bottleneck related to the precatalyst loading, which results from Rh^I^-sulfonate interactions, could be removed by “dilution” of the shell sulfonate functions with a neutral monomer, PEOMA. The resulting nanoreactors successfully catalyze the aqueous biphasic hydrogenation of styrene, but the performance is inferior to those of equivalent nanoreactors with neutral and polycationic shells. The lower activities observed during the recycles after the first run seem related to the catalyst alteration by migration towards the shell sulfonate functions. Hence, while the shell sulfonate dilution sufficiently suppresses the interaction with the “RhCl(COD)” fragment, and allows shell crossing during the precatalyst loading phase, the interaction with the activated “RhCl(TPP)_x_” functions during catalysis remains sufficiently strong to interfere with the operations of the active species. The higher catalyst leaching exhibited by these nanoreactors indicates their non-negligible transfer to the organic phase. This greater lipophilicity may result from the large fraction (80%) of PEO-containing monomers in the anionic shell. In all evidence, the presence of only 20% of charged sodium sulfonate groups does not impart sufficient lipophobicity to the nanoreactor shell. Further improvement of this catalytic tool requires the development of more lipophobic shells that are devoid of potentially coordinating functions, at least when mobile active species (i.e., anchored to the nanoreactor core *via* coordination bonds) are involved. In that respect, polycationic shells appear more promising than polyanionic ones. In addition, the synthetic protocol leading to nanoreactor assembly should be as simple and as straightforward as possible. Efforts in this direction continue in our laboratory.

## Data Availability

Findable, Accessible, Interoperable and Reusable (FAIR) data related to this publication will be deposited in the HAL repository (https://hal.archives-ouvertes.fr/hal-03852403, accessed on 1 October 2022) under the DOI of the present publication.

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
