# Peer review of "Phosphine-Functionalized Core-Crosslinked Micelles and Nanogels with an Anionic Poly(styrenesulfonate) Shell: Synthesis, Rhodium(I) Coordination and Aqueous Biphasic Hydrogenation Catalysis"

_polymers, 2022, doi:10.3390/polym14224937_

Round 1
Reviewer 1 Report
The manuscript entitled "Phosphine-functionalized core-crosslinked micelles and nanogels with an anionic poly(styrenesulfonate) shell: Synthesis, rhodium(I) coordination and aqueous biphasic hydrogenation catalysis" by H. Wang et al. was sent to the editorial office of the journal Polymers. This manuscript describes the synthesis and a series of tests for the resulting polymeric materials. The work undertaken is interesting and the results are promising. My main objection to this work is that the text is not concise. The text of articles should be as concise as possible. With such an elaborate text, it is hard to read and understand the content. In fact, every part of the manuscript is too long, starting with the abstract and introduction. Maybe only the experimental section is properly prepared. Figures should appear directly after mentioning them in the text. This problem applies to Figure 1. In many places in the manuscript, a word processor warning message is displayed. This applies to the indication of the corresponding figure in the supporting materials. Significantly, although the authors refer to the supporting materials, unfortunately they were not available for review. The Authors used different font colors in the text of the manuscript. It seems that this is not allowed except for supporting materials, drawings, etc. The Authors introduce many abbreviations. I think it would make it easier for the reader to understand the text if there was a separately made list of all the abbreviations used with their explanation. Of course, except for abbreviations that are understandable to a wide audience, for example, the abbreviation NMR does not need explanation. The experimental section regarding NMR measurements (l. 148-154) lacks information at what frequencies the signals for H-1 and P-31 were recorded. In l. 163 there is no reference to the software used. Software must also have its references. Not all drawings showing structural patterns have good resolution. It would be useful to improve the quality of Figures 1 and 8.
Author Response
We have tried to shorten the text as much as possible, removing minor details. There is, however, quite a lot of work described in this manuscript and we consider all material left in the main text as essential.
The figures have generally been placed immediately after the end of the paragraph where they are first called. In a few cases, however, this would generate large fraction of blank pages. In those cases, the figure has been placed in the next page, after the subsequent paragraph. The page editor will certainly improve this layout, whenever possible, prior to publication.
The word-processing error message was caused by hyperlinks to figures and tables in the Supporting Information. Once the manuscript and Supporting Information were separated, the hyperlinks ceased to work. We deeply apologize for not realizing this problem prior to submission. All these hyperlinks have now been transformed into plain text and there is no error remaining.
Concerning the font colors, we do believe that their use improves the readers’ ability to follow the arguments and facilitates understanding. Since color is allowed in images, why not in the text (if properly justified)? Since we did not receive specific instructions from the editor on this point, we have left this color coding, but the page editor may remove it, if necessary.
Concerning abbreviations, all uncommon acronyms have been defined the first time they are used. Certain journals have a separate section at the end of the manuscript for the definition of uncommon acronyms, but this is not the case for Polymers. We are ready to work with the Editor to further improve this point, if necessary.
The information of the NMR frequency is actually included in the instrument model: “All nuclear magnetic resonance spectra were recorded in 5 mm diameter tubes at 297 K on a Bruker Avance 400 spectrometer”. This means that protons are measured at 400 MHz and phosphorus at the corresponding frequency given by the P/H gyromagnetic ratios.
The problem of the figure quality (we believe the reviewer refers to figure 2 and 8, not 1 and 8) is not related to resolution, which is good, but rather to the small font size of the text in the figures. We have changed the layout of those two figures, allowing their expansion in size.
Reviewer 2 Report
The paper deals with the synergistic design, over a loop of improvement cycles as a consequence of structure activity relationships, of polymeric nanoparticles capable of supporting biphasic hydrogenation of styrene. The work is very interesting, the continuation of several studies of the lead author; it possesses a strong novelty character and clearly present noticeable advances over the state of the art. I suggest to accept this fine paper upon the authors adding the following relevant references: 10.1021/acsanm.2c02313; doi: 10.1007/s00289-012-0844-5
Author Response
We are grateful to this reviewer for the positive evaluation. We have introduced the citation of the first one of the two suggested contribution, which is a review article on single-chain nanoparticles, as the new reference [35]. The second suggested reference is related to a specific original research article with little relevance to the composition of our polymers, therefore we do not consider that its citation is essential.
Reviewer 3 Report
In this manuscript, the authors describe the synthesis, properties, and application in aqueous biphasic Rh-catalyzed hydrogenation of core-crosslinked micelle with a hydrophilic polyanionic outer shell and a phosphine-functionalized core. On the whole, the experimental process is performed well, and the quality of the experimental results and discussion is adequate and reasonable. So I suggest the acceptance to this work for publication on Polymers after minor revision. However, I think the authors should consider the following questions:
(1) With the increase of catalytic reaction time, the conversion rate gradually increases, indicating that the reaction molecules gradually contact the active site through diffusion. When the reaction time reaches 15 hours, the conversion rate reaches 100% (Figure 12). How long this 100% conversion rate can be maintained?
(2) As shown in Figure 13, the authors have carried out six cycle catalytic reactions, but the conversion rate obtained each time is not stable. What is the reason for this phenomenon? The author needs to give a reasonable explanation.
(3) As shown in Figure S28, the sizes of R0-[(SS-Na+)0.2-co-PEOMA0.8]x-b-St50-SC(S)SnPr are not uniform. The reviewer wants to know how to control the size uniformity based on the author's experience?
(4) What is the viscosity of the catalyst synthesized in this manuscript?
(5) These Rh-loaded latexes show good catalytic performance in the aqueous biphasic hydrogenation of neat styrene as a benchmark reaction. Meanwhile, the catalytic phase could be recovered and recycled. The reviewer wants to know whether this catalyst can be prepared on a large scale and how to preserve this type of catalyst?
(6) "Error! Reference source not found" appears many times in the manuscript, and the author needs to delete these statements.
Author Response
(1) We think that this reviewer did not correctly interpret the significance of Figure 12. The reaction is irreversible and Figure 12 shows a kinetic profile based on independent experiments run with different reaction times. Once converted, the substrate will not form again and the product is stable. Of course, the measurements are affected by experimental error and thus fluctuations may be observed, but that does mean that the product is converted back into the reactants. We do not consider that any change to the manuscript is needed.
(2) In this case, again, the reviewer may not have correctly interpreted the significance of the Figure. The observed variations of conversion vs. cycle number are not the result of “instability”. These are recycling experiments carried out with the same catalytic phase, each time recovered from the previous experiment and reused under the same conditions. Each conversion value, measured from the recovered organic phases after extraction, is the result of a different reaction carried out under the same conditions and, once again, the measured value is affected by experimental error. We believe that the meaning of these data and their description is already sufficiently clear from the current text and we do not consider that any change to the manuscript is needed.
(3) The uniformity of the self-assembled micelles mostly depends on the uniformity of the diblock chains that lead to the formations of these micelles. These particular micelles are rather unstable and heterogeneous because they are assembled from diblock copolymers with a short styrene block (50 units). More stable micelles with more uniform size resulted from diblocks with a longer (and better size-controlled) hydrophobic block.
(4) The reviewer probably means the viscosity of the latex containing the polymeric nanoreactors, which contain the catalyst. These are all very low-viscosity aqueous suspensions, as is typical for suspensions of spherical micelles. This is a well-known property, amply discussed in the literature. However, to help other readers who may not be familiar with this property, we have added the following sentence in the beginning of the Results and Discussion section: “The final latexes contain up to ca. 27% of polymer content in mass and have low viscosity, as expected for stable suspensions of spherical micelles.” We did not specifically measure the viscosity of our suspensions, which freely flow similarly to neat water.
(5) We do not see any problem in the potential scaling of the synthesis of these nanoreactor latexes. Our research is of fundamental nature and we do not have sufficient resources to scale up the synthesis, given the high cost of the rhodium catalyst. The stability is the same as for previously published nanoreactors with other types of hydrophilic shells. The following sentence has been added to the beginning of the Results and Discussion section: “The suspensions are stable, with no evidence of coagulation over several months, but must be stored under an inert atmosphere to avoid the slow aerial oxidation of the phosphine ligand to phosphine oxide.”
(6) As already explained in the reply to the comments of reviewer 1, this problem was caused by hyperlinks and has now been corrected.